# FarsightAlign TTS: Early-Stage Test-Time Scaling for Prompt-Aligned Text-to-Image Generation

## Abstract

Text-to-Image diffusion models have achieved remarkable progress under the guidance of "Scaling Laws", but further performance gains are increasingly hindered by diminishing returns from scaling model size and data volume. To bypass this bottleneck, Test-Time Scaling (TTS) has emerged as a promising alternative. However, the lack of interpretable signals in the early denoising steps forces existing TTS approaches to perform nearly complete denoising process for every candidate—resulting in high computational cost. In this work, we propose FarsightAlign TTS, a novel and efficient TTS method that leverages the rich semantic signals embedded in early cross-attention maps. With just a few denoising steps, FarsightAlign TTS can extract structured semantic information, such as object presence, layout, and attributes. It then leverages a lightweight scorer to prune unaligned candidates before committing to the final generation. This design significantly reduces computational cost while improving alignment with user's prompts. The experimental results demonstrate the effectiveness of our method. Furthermore, FarsightAlign TTS can function as a plug-and-play module, significantly boosting the semantic alignment capabilities of other advanced TTS frameworks with minimal additional computational overhead.

## 1 Introduction

The remarkable progress of Text-to-Image (T2I) diffusion models largely stems from "Scaling Laws"—enhancing performance by increasing model parameters and training data (Hoffmann et al., 2022; Kaplan et al., 2020). However, this approach encounters a significant bottleneck. As computational investment during training continues to grow, the improvements in model performance exhibit diminishing marginal returns. A fundamental reason for this trend is that merely increasing data and parameter counts is insufficient for models to acquire a deep understanding of complex logic, physical commonsense, and causal relationships. Consequently, even the largest models often fail to precisely align with nuanced human intent when faced with complex prompts (Agarwal et al., 2025; Mañas et al., 2024).

In contrast to allocating massive resources for marginal gains in training, users in practical applications have adopted a highly cost-effective strategy to overcome model limitations: generating multiple candidate images from the same prompt and manually selecting the best one. This widespread "Best-of-N" practice demonstrates that adding extra computation during inference time can significantly improve the final output quality (Brown et al., 2024). Inspired by this strategy, we shift our research focus from expensive model training to the more flexible and low-cost inference stage.

This perspective aligns with Test-Time Scaling (TTS) proposed in the domain of Large Language Models (LLMs) (Zhang et al., 2025a). In LLMs, to overcome the limitations of greedy decoding, researchers employ more complex search algorithms like beam search (Snell et al., 2024) or tree search (Xie et al., 2024) in test time. These algorithms actively explore a wider range of generation paths during inference, thereby producing higher-quality response with superior logical consistency and contextual coherence.

However, directly applying TTS to diffusion models is challenging. LLMs generate discrete tokens that can be decoded and evaluated at each step, whereas diffusion models operate in a high-

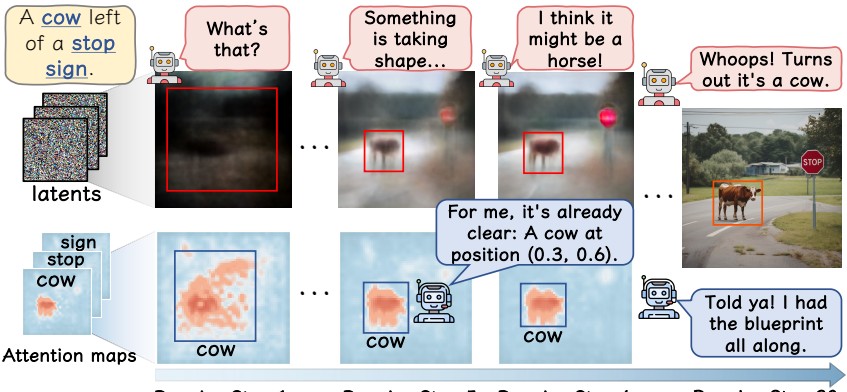

Figure 1: **Latent Decoding vs. Attention Maps.** In the early steps of diffusion, decoding intermediate latents (top row) produces perceptually blurry, semantically ambiguous images. In contrast, cross-attention maps (bottom row) already form a clear and accurate "blueprint" of the final scene, revealing precise object shape and location from the very beginning.

dimensional, non-interpretable latent noise space. The evaluation process for existing TTS methods in T2I tasks (Li et al., 2025b; Zhang et al., 2025c) requires decoding intermediate latents into full images. However, this presents a dilemma: decoding latents from early denoising stages yields blurry and uninformative images (as illustrated in Figure 1 and analyzed in Section 4.1), while decoding them from later stages incurs substantial computational cost due to the extensive number of denoising steps required. Furthermore, these methods focus only on general image quality, while overlooking whether the generated image truly aligns with the user's prompt (Agarwal et al., 2025).

These limitations demonstrate that an ideal TTS method requires **farsight**—the ability to discern a candidate's potential from the very early denoising steps. This capability can be leveraged to enhance T2I semantic **alignment**. Accordingly, we propose **FarsightAlign TTS**, a novel method built upon a key empirical observation: the cross-attention maps in the very early denoising steps already contain the critical semantic information that determines the final image composition (as shown in Figure 1).

FarsightAlign TTS operates by first sampling a large pool of initial noises. It then performs only a few denoising steps (just 5 steps) and, instead of costly decoding, directly extracts structured semantic features (object confidence, spatial location, attribute binding) from the early-stage cross-attention maps. A lightweight LLM scorer then rapidly prunes unpromising candidates based on their alignment with the prompt, allowing only the best to proceed to full generation.

Experimental results demonstrate that our method significantly outperforms existing TTS approaches on two mainstream benchmarks across three key evaluation metrics. The performance improvement is particularly pronounced in scenarios with limited computational resources.

Furthermore, the noise-filtering mechanism of FarsightAlign TTS is highly modular. We integrate this mechanism as a plug-and-play preprocessing module into other advanced diffusion TTS frameworks. Results show that with a negligible additional NFE cost, it can significantly boost their performance in semantic alignment.

To sum up, our main contributions are as follows:

- We propose FarsightAlign TTS, a computationally efficient TTS method that achieves strong semantic alignment with user prompts.

- As a plug-and-play module, FarsightAlign TTS seamlessly integrates with existing TTS methods, boosting performance with negligible computational overhead.

- Through extensive experiments, we prove that FarsightAlign TTS effectively utilizes test-time computation to consistently improve T2I performance, achieving a state-of-the-art comprehensive performance that surpasses all existing TTS methods.

## 2 RELATED WORKS

### 2.1 TEXT-TO-IMAGE DIFFUSION MODELS

In the field of T2I generation, early research primarily centered on GANs (Reed et al., 2016; Tao et al., 2022; Xu et al., 2018; Zhu et al., 2019) and autoregressive models (Ding et al., 2021; Gafni et al., 2022; Ramesh et al., 2021; Yu et al., 2022). As the scale of model parameters and training datasets has continued to expand, diffusion models have demonstrated powerful capabilities in image generation tasks (Balaji et al., 2022; Hoogeboom et al., 2023; Nichol et al., 2022; Ramesh et al., 2022; Saharia et al., 2022a). To mitigate the substantial computational costs associated with high-resolution image synthesis, Latent Diffusion Models have been introduced (Rombach et al., 2022). They enhance efficiency by performing the diffusion process within a low-dimensional latent space constructed by an autoencoder. For better alignment with the textual conditions, models like Stable Diffusion (Rombach et al., 2022; Podell et al., 2024; Esser et al., 2024) further incorporate cross-attention mechanisms (Chefer et al., 2023; Kim et al., 2025; Zhang et al., 2024b) to inject textual information. However, as we have highlighted in our introduction, despite their success, these models exhibit fundamental limitations in precisely follow complex prompts.

### 2.2 TEST-TIME SCALING

TTS improves model performance by leveraging extra computational resources during inference, typically through two approaches: 1) Generating diverse candidate outputs to increase the likelihood of high-quality results (Brown et al., 2024; Nguyen et al., 2024; Chen et al., 2024; Wang et al., 2023; Qiu et al., 2024; Zhang et al., 2025b), such as through Best-of-N sampling or self-consistency methods; 2) Extending reasoning depth, for example, using Chain-of-Thought prompting to guide step-by-step problem-solving for complex tasks (Wei et al., 2022; Madaan et al., 2023; Li et al., 2025a; Lightman et al., 2024; Xu et al., 2025). To further refine the outputs, researchers often combine intermediate-reward models (Yao et al., 2023; Zhang et al., 2024a; Xie et al., 2023) with search strategies (Xie et al., 2024; Zhou et al., 2024; Deng & Raffel, 2023) to dynamically guide and select among generation paths. However, due to the differences in architecture and generation mechanisms between diffusion models and LLMs, directly migrating TTS methods from LLM to T2I generation poses significant challenges.

### 2.3 TTS FOR T2I MODEL

A dominant implementation of TTS for T2I models is search-based guidance. This class of methods frames the multi-step denoising process as a decision tree, where each node represents an intermediate latent state, and employs search algorithms to explore the optimal generation path. For instance, researchers have successfully applied strategies like Dynamic Search (Li et al., 2025b), classic algorithms such as A* (Zhang et al., 2025c), and evolutionary algorithms (He et al., 2025). The core idea is to periodically assess the quality of intermediate nodes and prune low-quality branches, thereby focusing computation on more promising paths. Nevertheless, these approaches are computationally expensive due to the need to decode intermediate latents, and they also struggle to achieve precise semantic alignment with user prompts.

## 3 PRELIMINARIES

### 3.1 DENOISING PROCESS IN LATENT DIFFUSION MODELS

Diffusion models generate images by iteratively denoising a random latent vector, $z_T \sim \mathcal{N}(0, I)$. At each step $t$, a network $\epsilon_\theta$ predicts the noise $\epsilon_\theta(z_t, t, c)$ conditioned on a text prompt $c$. A scheduler then computes the preceding latent $z_{t-1}$ using this prediction:

$$z_{t-1} = \mathcal{S}(z_t, \epsilon_\theta(z_t, t, c)). \tag{1}$$

This process repeats until the final latent $z_0$ is reached, which a decoder $\mathcal{D}$ maps to the final image $x_0 = \mathcal{D}(z_0)$.

## 3.2 CROSS-ATTENTION MAPS

Cross-attention layers align image generation with a text prompt by treating intermediate image features $\phi(z_t)$ as the Query ($Q$), and text embeddings $\tau(c)$ as the Key ($K$) and Value ($V$). At a timestep $t$, the attention matrix $A$ is calculated as:

$$A = \text{softmax}\left(\frac{QK^T}{\sqrt{d_k}}\right). \tag{2}$$

Each column of $A$ is an attention map for a specific text token. For our method, we create a single, unified attention map per token by extracting its map from all cross-attention layers, upsampling each to $64 \times 64$, and averaging the results. This aggregated map represents the token's total spatial influence and is the key signal we leverage for early-stage evaluation.

## 3.3 TEST-TIME SCALING

To extend the performance limits of the model in test time, a common strategy is to generate $N$ candidates in parallel for a single prompt $c$. This is achieved by initializing a batch of independent latent vectors, $\{z_T^i\}_{i=1}^N$, to launch $N$ parallel generation trajectories.

A reward function, $R(\cdot)$, is used to evaluate the quality of each trajectory. This evaluation can target the final image $x_0^i$, or intermediate states during the denoising process, enabling assessment and guidance of the generation.

The final output is the image $x_0^{i^*}$ from the trajectory with the highest cumulative reward score, $S_i$. The optimal index $i^*$ is found by:

$$i^* = \underset{i \in \{1,\ldots,N\}}{\arg\max}\ S_i. \tag{3}$$

# 4 METHOD

## 4.1 ATTENTION MAPS VS. DECODED LATENTS

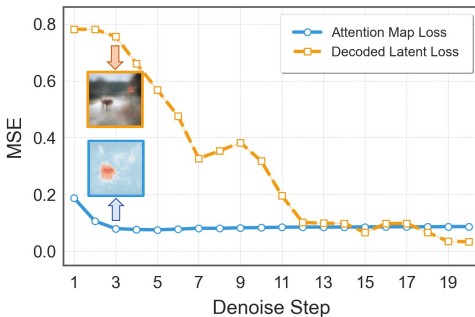

Figure 2: **Attention Map Loss vs. Decoded Latent Loss.** Attention maps (blue) provide a stable, low-error signal for object position throughout the denoising process, unlike the initially high and unstable decoded latent loss (orange).

A pivotal challenge for TTS in diffusion models lies in identifying a reliable signal at a very early denoising stage to avoid the high cost of a full denoising process for each candidate. A prevalent approach relies on decoding intermediate latents $z_t$ into images $x_t$ and then using reward models for evaluation. However, the critical flaw in this method is that early-stage decoded images are perceptually blurry and semantically ambiguous, as illustrated in Figure 1 (Top). This ambiguity prevents reward models from reliably assessing alignment with the text prompt, making the evaluation signal unreliable.

To address this, we propose using cross-attention maps directly as the evaluation proxy. We find that due to the inherent coupling between text tokens and their spatial layout, these maps form a clear semantic blueprint of the final image's object composition from the earliest denoising stages, as shown in Figure 1 (Bottom). This provides a far more stable and informative early signal.

To quantitatively compare these two proxies, we measure their positional error against a ground-truth position, which we obtain from the final image $D(z_0)$. We define $M_{\text{det}}(\cdot)$ as an operator that returns the center coordinate of the detected object's bounding box.

The Decode Loss, $\mathcal{L}_{\text{decode}}(t)$, is the Euclidean distance between the position predicted from the intermediate image $D(z_t)$ and the ground truth. The Attention Map Loss, $\mathcal{L}_{\text{attn}}(t)$, similarly measures the distance from the attention map's centroid to the ground truth. The losses are formulated as:

$$\mathcal{L}_{\text{decode}}(t) = \|M_{\text{det}}(D(z_t)) - M_{\text{det}}(D(z_0))\|_2 \tag{4}$$

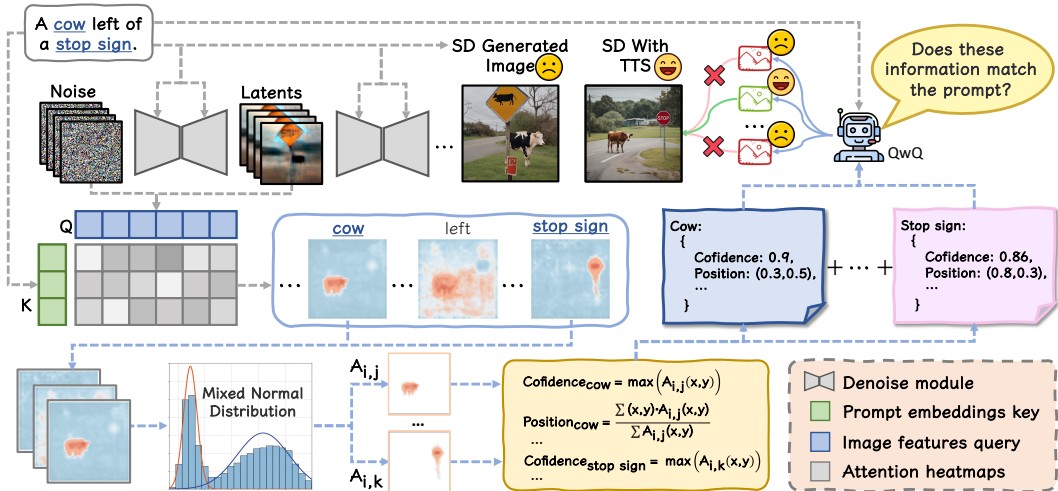

Figure 3: **An overview of FarsightAlign TTS.** To efficiently improve generation quality, we initiate multiple parallel candidates. Instead of costly full decoding for each candidate, we analyze their cross-attention maps at an early denoising stage. From these maps, we extract key semantic information including object confidence, position, and attribute binding scores. This information is used to score each candidate's alignment with the prompt, allowing us to select the highest-scoring candidate for the full generation process while discarding the rest.

$$\mathcal{L}_{\text{attn}}(t) = \|\text{Centroid}(A_t) - M_{\text{det}}(D(z_0))\|_2 \tag{5}$$

As shown in Figure 2, the results are definitive. $\mathcal{L}_{\text{attn}}(t)$ (blue line) is minimal and stable from the outset, whereas $\mathcal{L}_{\text{decode}}(t)$ (orange line) starts at a high value and diminishes slowly through the early denoising step. This stark divergence confirms that attention maps are a far superior early-stage proxy. This finding is the cornerstone of our method, enabling the efficient selection of an optimal initial latent $z_T$ by sidestepping the need for costly denoising steps.

## 4.2 FARSIGHTALIGN TTS

Similar to other TTS approaches, we first initialize a pool of $N$ initial noises for a given text prompt. After applying a few denoising steps to each, we extract the corresponding batch of attention maps, which are then processed in parallel to distill key semantic information.

### 4.2.1 ADAPTIVE OBJECT EXTRACTION FROM ATTENTION MAPS

A straightforward approach to segmenting attention maps would be to apply a simple threshold. However, we find this method to be unreliable, as the optimal threshold—whether a fixed value or a percentile—varies dramatically across different tokens and denoising steps. To address this, we propose an adaptive method that requires no manually-tuned hyperparameters.

Our approach is based on the observation that the intensity distribution of an attention map typically exhibits a distinct bimodal structure, as shown in Figure 3. This bimodality is a natural consequence of the cross-attention mechanism; dot-product similarity scores, when transformed by the softmax function, inherently partition pixels into high-activation (object) and low-activation (background) groups.

Leveraging this property, we propose to explicitly model this structure by decomposing the distribution into two Normal Distribution representing the object and background, respectively. The parameters for this mixture are estimated via the Expectation-Maximization, providing a principled method for pixel-level segmentation. The probability density function for a pixel intensity $x_i$ is given by:

$$p(x_i \mid \theta) = \alpha \cdot \mathcal{N}(x_i \mid \mu_o, \sigma_o^2) + (1 - \alpha) \cdot \mathcal{N}(x_i \mid \mu_b, \sigma_b^2) \tag{6}$$

here,$(\mu_o, \sigma_o^2)$ and $(\mu_b, \sigma_b^2)$ are the mean and variance for the high-activation object and low-activation background components, respectively, and $\alpha$ is the object's mixture proportion. Upon

fitting the model, each pixel is assigned to the component—object or background—with the higher posterior probability. This results in a binary mask $\mathcal{R}_{i,j}$, representing the extracted object region for token $j$ in candidate $i$.

Appendix A.2.3 provides a detailed performance comparison against fixed-thresholding segmentation approaches.

### 4.2.2 OBJECT GENERATION CONFIDENCE

With the object region $\mathcal{R}_{i,j}$ extracted, we first quantify its generation confidence. A well-generated object corresponding to token $j$ should manifest as a concentrated region of high activation in its attention map (validated in Appendix A.2.1). We therefore define the object's generation confidence as the peak intensity value within its object region $\mathcal{R}_{i,j}$. This provides a direct measure of how strongly the model focuses on the object. The confidence score is given by:

$$P_{i,j} = \max_{(x,y)\in\mathcal{R}_{i,j}} A_{i,j}(x,y) \tag{7}$$

where $P_{i,j}$ is the generation confidence for token $j$ in candidate $i$, and $A_{i,j}(x,y)$ is the attention intensity at pixel coordinate $(x,y)$.

### 4.2.3 POSITIONAL INFORMATION

Beyond confidence, we determine the object's location by computing the center of mass (centroid) of the high-activation region $\mathcal{R}_{i,j}$. This is a weighted average of the pixel coordinates, where the weights are the attention intensities, providing a robust estimate of the object's spatial position. The centroid coordinates $(\mathrm{Pos}^i_{x,j}, \mathrm{Pos}^i_{y,j})$ are computed as:

$$\mathrm{Pos}^i_{x,j} = \frac{\sum_{(x,y)\in\mathcal{R}_{i,j}} x \cdot A_{i,j}(x,y)}{\sum_{(x,y)\in\mathcal{R}_{i,j}} A_{i,j}(x,y)}, \mathrm{Pos}^i_{y,j} = \frac{\sum_{(x,y)\in\mathcal{R}_{i,j}} y \cdot A_{i,j}(x,y)}{\sum_{(x,y)\in\mathcal{R}_{i,j}} A_{i,j}(x,y)}, \tag{8}$$

where the summation is performed over all pixels within the high-activation region $\mathcal{R}_{i,j}$.

### 4.2.4 ATTRIBUTE BINDING

Quantifying the binding between an object $o$ and an adjective $a$ requires measuring how precisely the attribute's attention is focused on the object. Naive overlap metrics are insufficient as they ignore attention intensity. We therefore introduce a directional binding score, $B_{a\rightarrow o}$, defined as the mean attention intensity of the adjective's map, $A_a$, over the object's pre-segmented region, $\mathcal{R}_o$. Formally, this is expressed as:

$$B_{a\rightarrow o} = \frac{1}{|\mathcal{R}_o|} \sum_{(x,y)\in\mathcal{R}_o} A_a(x,y) \tag{9}$$

where $|\mathcal{R}_o|$ is the area of the object's region. A higher $B_{a\rightarrow o}$ score signifies a stronger semantic binding, indicating that the attribute is more precisely associated with the object.

### 4.2.5 LLM-GUIDED CANDIDATE SELECTION

For each of the $N$ initial candidates, we aggregate the extracted semantic metrics—object confidence, position, and attribute binding—into a structured JSON object. This creates a batch of $N$ JSONs, each serving as a compact semantic summary of a potential outcome (see Figure 3 for an example).

This collection of semantic summaries is then scored by an LLM acting as a high-level semantic judge. We employ batch-processing to optimize for latency; the LLM evaluates a batch of candidate summaries against the original user prompt in each forward pass, assigning a holistic alignment score to each. For a detailed description of the LLM scorer, please refer to Appendix A.1.4.

Finally, the candidate with the highest score is selected for the full denoising process, while the remaining $N-1$ candidates are discarded.

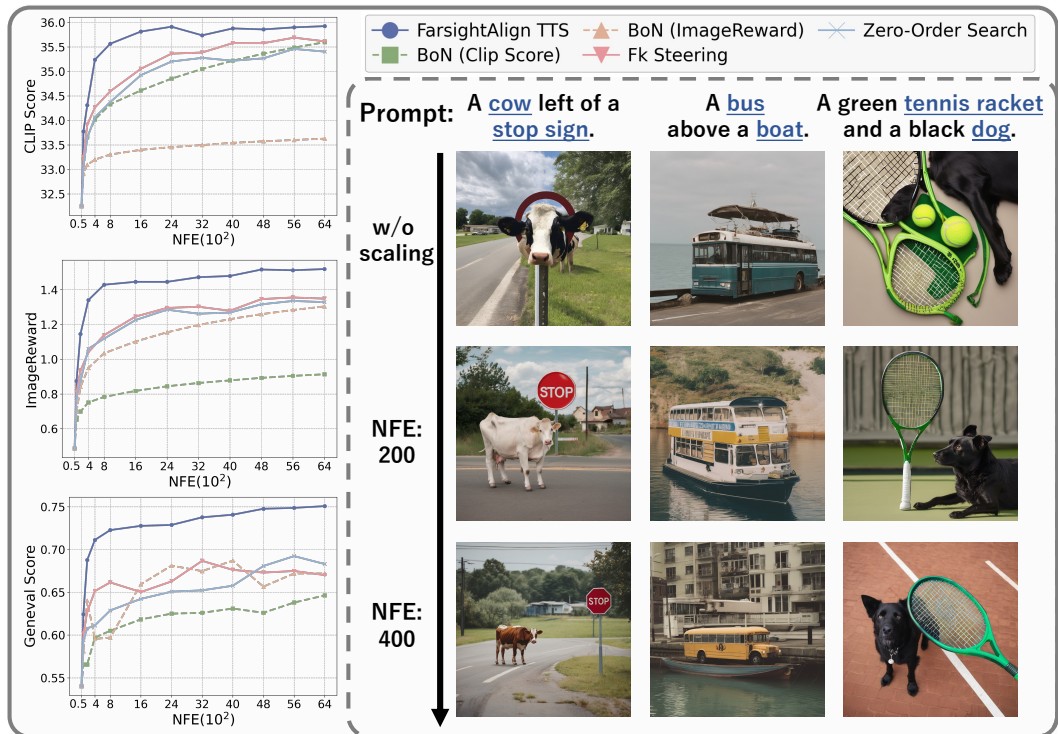

Figure 4: **FarsightAlign TTS outperforms baselines on the Geneval.** The plots (left) show superior quantitative performance across three metrics, while the images (right) demonstrate improved adherence to complex prompts. **See Appendix A.3 for additional qualitative comparisons.**

## 5 EXPERIMENTS

### 5.1 EXPERIMENT SETUP

Our tests are performed on two mainstream benchmarks, Geneval Benchmark (Ghosh et al., 2023) and DrawBench (Saharia et al., 2022b), using two distinct model architectures: the U-Net-based SDXL (Podell et al., 2024) and DiT-based SD3 (Esser et al., 2024). For both models, we use 50 denoising steps with other hyperparameters remaining as the default.

We compare our method against three representative TTS baselines. The first is Best-of-N (BoN), a straightforward approach that selects the best image from a pool of fully generated candidates via a reward model. The second is FK Steering (Singhal et al., 2025), a particle-sampling method that iteratively refines a set of candidates throughout the denoising process. The third is Zero-Order Search (Z-O Search) (Ma et al., 2025), an algorithm that finds an optimal solution through iterative neighborhood sampling and evaluation. Finally, to ensure an absolutely fair assessment, the metrics used for our final performance measurement—including Geneval Score, ImageReward Score (Xu et al., 2023), and Clip Score (Hessel et al., 2021)—are not integrated into FarsightAlign TTS's reward function.

### 5.2 RESULT ANALYSIS

#### 5.2.1 COMPARATIVE ADVANTAGES OVER EXISTING TTS METHODS

We first evaluate FarsightAlign TTS on the Geneval benchmark, measuring its test-time computation by the Number of Function Evaluations (NFE).

As illustrated in Figure 4, our method's performance consistently improves with increasing NFE, and outperforming other TTS methods. By eliminating unpromising candidates early in the denoising process, our method avoids wasting NFE on low-quality generation paths. This advantage is most significant in low-NFE scenarios, which align with practical usage constraints.

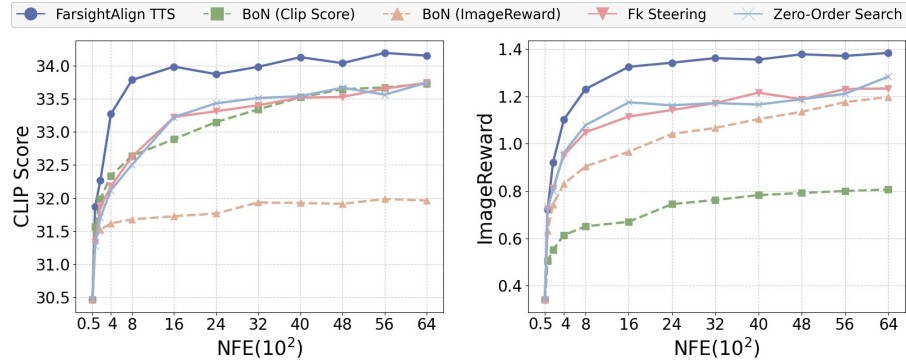

Figure 5: **Performance comparison on the DrawBench.** We also conduct experiments on the DrawBench using CLIP Score (left) and ImageReward (right).

Furthermore, we observe a critical phenomenon: while many existing TTS methods achieve stable gains on metrics they are directly optimized for (e.g., ClipScore and ImageReward), they exhibit significant performance volatility on the unseen metric: Geneval Score. This strongly suggests that these methods may be overfitting to specific reward models, failing to genuinely enhance the T2I model's ability to generate images that align with human semantics.

To further validate the generalization capabilities of our approach, we also conduct experiments on the DrawBench, using Clip Score and ImageReward as metrics. The results, illustrated in Figure 5, reaffirm that FarsightAlign TTS maintains a significant performance lead, demonstrating its robustness across different benchmarks.

### 5.2.2 ANALYSIS OF EFFICIENCY

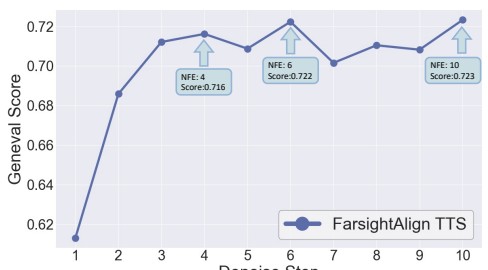

The core of our method's efficiency lies in its ability to extract informative attention maps at a very early stage of the denoising process. To quantify this advantage and identify the optimal pruning point, we conduct an ablation study. Under a fixed total computational budget of 400 NFE, we evaluate the final performance when using attention maps extracted at different denoising steps (from 1 to 10) for pruning.

Figure 6: **Relationship between the Geneval Score and the denoising step chosen for pruning.** Performance gains saturate after just 4 steps.

As shown in Figure 6, performance gains saturate after just 4 steps, indicating that we can effectively screen a candidate for only 4 NFE. In stark contrast, conventional TTS methods require the full generation process ( 50 NFE) to evaluate a single candidate, making our approach nearly 13 times more cost-effective. This drastic efficiency gap allows our method to explore a vastly larger search space of initial noises within the same NFE budget.

### 5.2.3 SYNERGY AS A PLUG-AND-PLAY MODULE

Existing TTS methods often struggle in low-NFE scenarios, as the initial random sample pool may lack high-potential candidates. Leveraging the insight that initial noise critically shapes the final image (Mao et al., 2024), we propose FarsightAlign TTS as a plug-and-play pre-filtering module. It efficiently prunes a large set of initial noises at a minimal computational cost, providing a high-quality subset of candidates for a subsequent TTS search algorithm. This pre-filtering ensures the search begins with semantically aligned candidates, preventing wasted computation on poor initializations (see Appendix A.1.3 for implementation details).

To validate this synergy, we integrated our module with several baseline TTS methods. As shown in Figure 7, this integration substantially boosts performance on the Geneval benchmark. The augmented methods show the most significant gains in the low-NFE regime, confirming our approach's effectiveness.

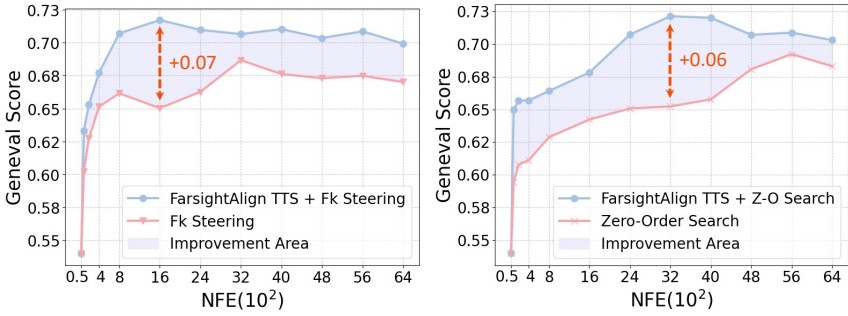

Figure 7: **Synergy with existing Test-Time Sampling methods.** FarsightAlign TTS significantly improves the Geneval Score of Fk Steering and Zero-Order Search, particularly at lower NFE.

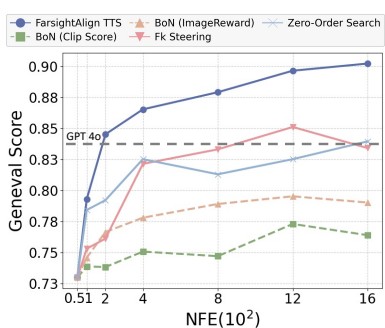

Figure 8: **Performance comparison on SD3.** Our method continues to significantly outperform all baselines on the Geneval benchmark.

Figure 9: **Comparison of generation diversity.** The bar chart (left) provides a quantitative diversity score (based on the L2 distance of CLIP features), while the right panel shows a qualitative comparison of generated samples.

### 5.2.4 GENERALIZABILITY ACROSS ARCHITECTURES

To further validate the architecture-agnostic nature and generalization capability of our method, we extend its application from the U-Net-based SDXL to the SD3, which is built upon DiT architecture. As shown in Figure 8, FarsightAlign TTS demonstrates superior performance on the Geneval benchmark with SD3 as well. Notably, with a computational budget of just 200 NFE (equivalent to generating 4 images), its performance even surpasses the powerful closed-source model, GPT-4o (OpenAI, 2023).

### 5.2.5 MAINTAINING GENERATION DIVERSITY

A significant pitfall of TTS methods is "reward hacking," where over-reliance on reward models leads to reduced diversity (Ma et al., 2025). We evaluate this on the Geneval benchmark by measuring the diversity of 8 generated images (400 NFE budget) per prompt, quantified as the average L2 distance between their CLIP features. As shown in Figure 9, FarsightAlign TTS maintains significantly higher diversity than its competitors. We attribute this success to our pruning strategy, which relies on fundamental semantic alignment rather than external, potentially biased reward models that can hinder diversity.

## 6 CONCLUSION

We propose FarsightAlign TTS, a test-time scaling method that improves semantic alignment in Text-to-Image diffusion models. Instead of relying on costly full-image decoding, our approach extracts semantic cues—such as object presence and layout—from early-stage cross-attention maps, allowing efficient candidate pruning via a lightweight LLM scorer.

Through extensive experiments, we demonstrate that FarsightAlign TTS consistently surpasses existing TTS methods across multiple benchmarks, especially in low-computation settings. As a modular plug-in, it also enhances other TTS frameworks with minimal overhead.

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

# A  APPENDIX

## APPENDIX DIRECTORY

## A.1  EXPERIMENTAL SETUP AND IMPLEMENTATION

### A.1.1  DATASETS

We provide further details on the benchmarks used for our evaluation.

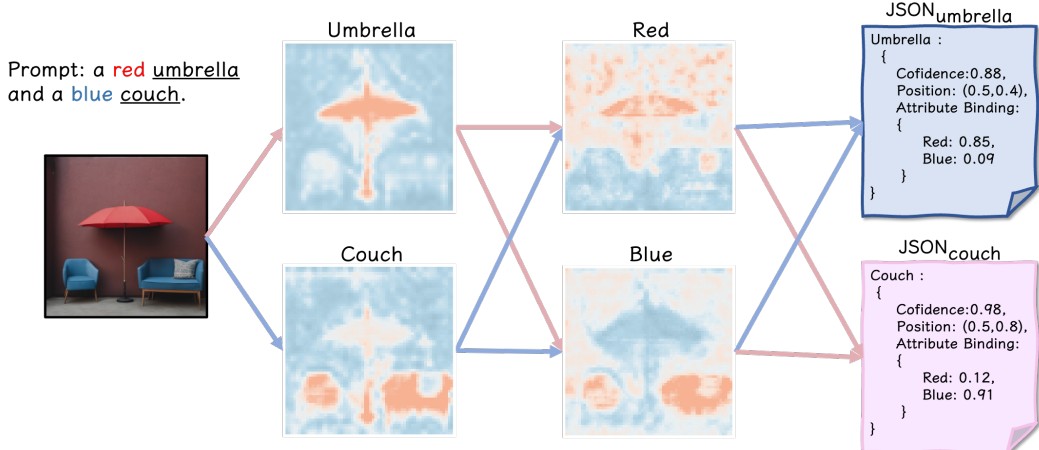

Figure 10: Information Extraction Pipeline. An illustration of how FarsightAlign processes cross-attention maps to produce a structured JSON summary. This summary quantifies object confidence, position, and attribute bindings, forming the basis for our efficient, early-stage candidate pruning.

Geneval Benchmark: An object-centric framework to evaluate T2I models on complex prompts. It contains 500 prompts across four categories: attribute binding, object relationships, spatial reasoning, and multi-object composition. Geneval provides a more granular, instance-level analysis than holistic metrics like FID, helping to identify specific model failure modes.

DrawBench: A diverse benchmark with 200 challenging prompts designed to probe the limits of T2I models. It specifically tests difficult aspects like compositionality, negation, and unusual object interactions.

### A.1.2 EVALUATION METRICS

We employed three metrics for a multi-dimensional comparison. For a fair evaluation, none were used to guide the FarsightAlign TTS process.

CLIP Score: Measures semantic consistency between the prompt and the image using the cosine similarity of CLIP embeddings (ViT-B/32).

ImageReward: A reward model trained on human preferences to evaluate overall quality, including image aesthetics, realism, and text-to-image alignment.

Geneval Score: The native, object-centric eva luation suite for the Geneval benchmark. It performs fine-grained, automated verification of semantic elements (e.g., presence, attributes, position) using a vision-based pipeline. This provides a strict measure of prompt fidelity, less susceptible to reward hacking than holistic scores.

### A.1.3 PLUG-AND-PLAY PIPELINE IMPLEMENTATION

In this section, we provide a detailed description of the implementation for integrating FarsightAlign TTS as a plug-and-play module. We outline the two-stage pipeline that enables FarsightAlign TTS to act as an efficient pre-filtering stage, thereby optimizing the initial candidate pool for subsequent complex search algorithms like FK Steering and Zero-Order Search.

**Stage 1: FarsightAlign Pre-filtering.** The process begins by sampling a large pool of $8 * N$ initial noise latents $\{z_{T,i}\}_{i=1}^{8*N}$. Instead of a full denoising process, we perform only a small number of denoising steps, $T_{\text{prune}}$ (typically 5 steps), for all $8 * N$ candidates. At step $T - T_{\text{prune}}$, we extract the cross-attention maps for each candidate and use our scorer to evaluate its semantic alignment with the prompt. We then rank all candidates by their scores and select the initial noise latents of the top-$N$ candidates, forming a high-potential subset $\{z_{T,j}^*\}_{j=1}^{N}$.

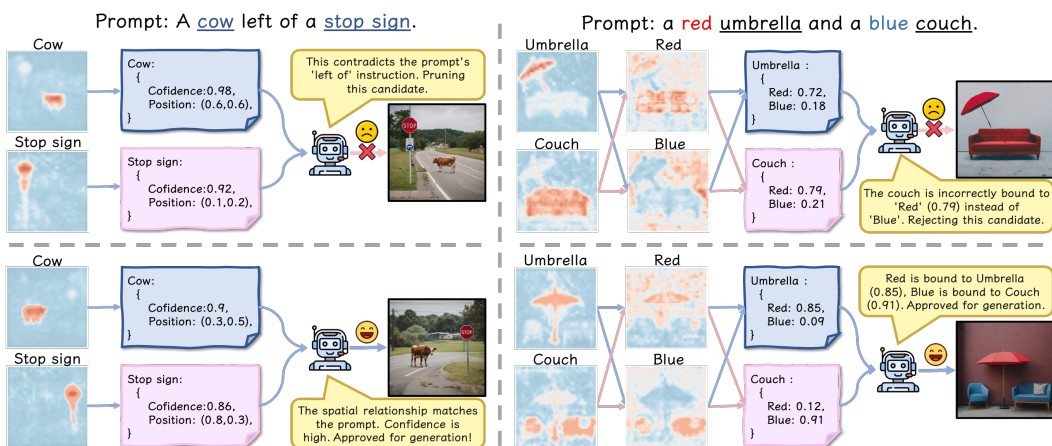

Figure 11: Early-Stage Pruning Mechanism. Our method successfully identifies and prunes semantically flawed candidates at an early stage by reasoning over structured data derived from attention maps. It correctly rejects candidates with incorrect spatial layouts (left) and mismatched attribute bindings (right), while approving logically sound ones.

**Stage 2: Main TTS Method Execution.** This pre-screened, high-potential subset of $N$ initial latents is then passed as the starting candidate pool to the main TTS method (e.g., FK Steering or Z-O Search). The main method then executes its full, more computationally intensive algorithm—be it iterative particle refinement or neighborhood search—but its exploration is now confined to this semantically strong set of candidates.

### A.1.4 LLM SCORER IMPLEMENTATION

The LLM scorer is the high-level reasoning engine of FarsightAlign TTS. Its primary role is to interpret the structured semantic data extracted from early-stage attention maps and evaluate its alignment with the complex, compositional intent of the user's prompt.

For each candidate, the extracted metrics—object confidence, position, and attribute binding scores—are aggregated into a structured JSON object. This "semantic summary", as illustrated in Figure 10, provides a compact, machine-readable representation of the potential image.

To optimize resource consumption and minimize latency, we aggregate the summaries from multiple candidates into a single batch and feed them to the LLM in one forward pass. The model then returns a corresponding batch of evaluation scores.

To balance high throughput with model performance, we process the semantic summaries in batches. A primary constraint is the context length of the LLM, as excessively long inputs can degrade reasoning quality. We therefore set our batch size to 64. This choice is calibrated based on the capabilities of powerful models such as QwQ, which have been shown to maintain excellent performance even with context lengths approaching 8192 tokens (Team, 2025). A batch of 64 JSON summaries (totaling approximately 6000 tokens) remains comfortably within this high-performance range, ensuring both processing efficiency and reliable evaluation.

To elicit the reasoning shown in our examples (as shown in Figure 11), we use the specific zero-shot system prompt detailed in Figure 12. This template, which defines the LLM's role, evaluation logic, and required output format, is applied to each candidate's summary within the batch.

### A.2 ABLATION STUDIES AND COMPONENT ANALYSIS

### A.2.1 VALIDATION OF OBJECT GENERATION CONFIDENCE

This section provides the visual evidence for our core claim in Section 4.2.2: that the peak intensity of an early-stage attention map is a highly reliable predictor of whether a corresponding object will be successfully generated.

```
You are an expert evaluator for a Text-to-Image
model. Your task is to evaluate a LIST of
candidate semantic summaries against a single
user prompt. For each summary in the list, you
must evaluate it across three distinct criteria,
providing a score from 1 to 10 for each.

Your evaluation criteria for EACH candidate are:
1. Object Presence: Are all key objects present
   with high confidence?
2. Spatial Relationships: Are spatial instructions
   (e.g., 'left of') correctly represented?
3. Attribute Bindings: Are attributes (e.g., 'red')
   correctly assigned to the right objects?

USER PROMPT:
{user_prompt}

LIST OF CANDIDATE SUMMARIES (JSON ARRAY):
{list_of_json_summaries}

Based on your evaluation, provide a JSON array.
Each object in the array should contain the three
scores for the corresponding candidate summary.
The order of objects in your output array MUST
match the order of the summaries in the input
array. Your output must be only the JSON array.

Example Output (for a batch of 2):
[
  {
     "object_presence_score": 8,
     "spatial_relations_score": 9,
     "attribute_binding_score": 10
  },
  {
     "object_presence_score": 10,
     "spatial_relations_score": 2,
     "attribute_binding_score": 7
  }
]
JSON ARRAY OUTPUT:
```

Figure 12: The prompt template for batched, factorized scoring. It instructs the LLM to process a list of candidate summaries and return a corresponding list of JSON objects containing the evaluation scores, ensuring high throughput and parsable output.

Figure 13 demonstrates this direct correlation by contrasting two sets of outcomes. The key distinction is not merely the quality of the final image, but whether the prompted object is present or absent—a factor dictated by the strength of its attention map.

- **Top Row (Low Peak Intensity → Failed Generation):** This row illustrates cases where the model failed to generate a key object. For instance, the relevant concepts required by the prompt (e.g., "cow" or "dog") are malformed, incoherent, or entirely absent in the final images. Their corresponding attention maps are visibly **diffuse, scattered, and lack any distinct high-intensity peak**. This weak, unfocused attention signals the model's failure to ground the concept, directly leading to the object's failed generation in the final output.

- **Bottom Row (High Peak Intensity → Successful Generation):** In stark contrast, this row showcases successful generations where all prompted objects are accurately rendered. The attention maps corresponding to these objects—such as the "cow" or "dog"—exhibit a **sharp, localized, and high-intensity peak**. This focused activation indicates that the model has confidently materialized the concept onto the canvas, directly resulting in a well-formed and clearly recognizable object.

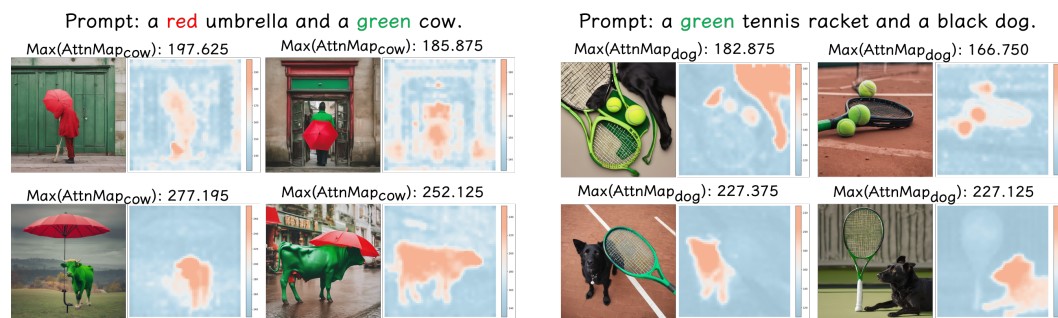

Figure 13: **Visual Evidence for FarsightAlign's Confidence Metric.** A comparison showing that successfully generated objects (bottom row) correspond to early-stage attention maps with high-intensity, focused peaks, while failed generations (top row) correspond to diffuse, low-intensity maps.

| Method | Geneval Score (%) |
|---|---|
| w/o TTS | 20.03 |
| **BoN (Clip Score)** | **28.99** |
| **BoN (Peak Intensity)** | **28.90** |

Table 1: **Performance on GenEval Benchmark (Color Attribution).** Comparison of our BoN approach using peak intensity as a reward signal against a baseline model.

Comparing the attention peaks of the top and bottom rows reveals that the peaks for the successfully generated targets in the bottom row are significantly higher than those in the top row.

Furthermore, we conducted a supplementary study to further validate the efficacy of attention peak intensity as a predictor of generation success. In this experiment, we employed peak intensity as a reward signal within a BoN sampling strategy.

For each prompt, we generated 32 candidate samples. We then calculated the early-stage attention peak intensity for the specified object in each sample and selected the image with the highest peak intensity as the final output. This methodology was evaluated on the GenEval benchmark, with a particular focus on the "Color Attribution" category of prompts, which are notoriously prone to object omission.

As shown in Table 1, our method demonstrates a significant improvement on the GenEval benchmark (Color Attribution), achieving performance comparable to BoN guided by CLIP scores. Critically, our approach provides this competitive performance without requiring the computationally expensive step of full image denoising for candidate selection.

In summary, these experiments demonstrate that early-stage attention peak intensity is a powerful and efficient metric. The visual evidence confirms it is a reliable predictor of successful object generation. Furthermore, when applied as a reward signal, it becomes a computationally cheap optimization tool that improves output quality without the need to fully generate every candidate image, unlike more costly methods such as CLIP-based scoring.

A.2.2 ABLATION ON SCORER CHOICE

To determine the optimal LLM scorer for our framework, we conducted an ablation study to evaluate the impact of different models on final generation quality.

The study was performed under a fixed computational budget of NFE=400 for the generation process, ensuring a fair comparison where the primary variable was the scorer model. We evaluated four distinct models:

- QwQ-32B (Full Precision)
- QwQ-32B (8-bit Quantized)

| Scorer Model | Geneval Score |
|---|---|
| Qwen2.5-7B | 0.685 |
| Qwen3-30B-A3B | 0.723 |
| QwQ-32B (Full Prec.) | 0.720 |
| QwQ-32B (8-bit Quant.) | 0.716 |

Table 2: Ablation study on scorer model choice under a fixed NFE=400 budget. Performance is measured by the final Geneval Score. The 8-bit quantized QwQ-32B provides the best balance of performance and efficiency.

- Qwen2.5-7B

- Qwen3-30B-A3B

The results, summarized in Table 2, reveal several key insights. Interestingly, most of the larger models (QwQ-32B, QwQ-32B 8-bit, and Qwen3-30B-A3B) yielded comparable final generation quality, with only marginal differences in performance metrics. The smallest model, Qwen2.5-7B, performed slightly worse, suggesting that a certain threshold of reasoning capability is necessary for the task.

Crucially, the 8-bit quantized version of QwQ-32B achieved performance nearly identical to its full-precision counterpart. This demonstrates that for our specific task of evaluating structured JSON data, the quantization process does not lead to a significant degradation in the model's critical reasoning abilities. Based on this analysis, we selected the 8-bit quantized QwQ-32B as the final scorer.

### A.2.3 ANALYSIS OF ADAPTIVE OBJECT EXTRACTION

As stated in the main paper (Section 4.2), a core component of our framework is the ability to reliably segment an object's region from its cross-attention map. This section provides the detailed performance comparison that justifies our choice of an adaptive segmentation method over simpler thresholding techniques.

To comprehensively evaluate the quality of our adaptive segmentation method, we designed two distinct experiments. The first assesses the positional accuracy of the extracted masks at an early stage, while the second measures the downstream impact of the segmentation method on the final image generation quality.

**Experiment 1: Positional Accuracy of Extracted Objects** A high-quality segmentation should accurately predict the final position of an object, even from an early denoising step. To measure this, we designed the following evaluation:

1. At an early denoising step (5 step), we use each segmentation method to extract the object mask and compute its geometric center (centroid).

2. After the full denoising process is complete, we run a object detector on the final generated image to obtain the ground-truth bounding box for the object.

3. We then calculate the Mean Positional Error, defined as the average Euclidean distance between the predicted centroid from the early stage and the center of the final ground-truth bounding box.

The results, averaged over the Geneval benchmark, are presented in Table 3. A lower error indicates that the segmentation method provides a more accurate and stable localization signal early on. Our adaptive method achieves a significantly lower positional error, demonstrating its superior ability to pinpoint the object's final location.

**Experiment 2: Impact on Final Generation Quality** To assess how segmentation quality affects the end-to-end performance of our framework, we conducted a direct comparison. We integrated both the Percentile Thresholding and our Adaptive method into the full FarsightAlign pipeline and evaluated their final output on the Geneval benchmark under a fixed computational budget of NFE=400.

| Segmentation Method | Mean Positional Error |
|---|---|
| Fixed Threshold (T=200) | 0.38 |
| Percentile Threshold (P=90%) | 0.15 |
| **Ours** | **0.07** |

Table 3: Positional accuracy of different segmentation methods. The error is the L2 distance between the centroid predicted at an early stage and the final object center. Lower is better.

| Segmentation Method | Geneval Score |
|---|---|
| Percentile Threshold (P=90%) | 0.697 |
| **Ours** | **0.716** |

Table 4: End-to-end performance of FarsightAlign using different segmentation modules under a fixed NFE=400 budget. Higher is better.

As shown in Table 4, the choice of segmentation method has a direct and notable impact on the final result. By providing more accurate semantic information to the LLM scorer, our adaptive approach leads to a clear improvement in the final Geneval Score.

## A.3 QUALITATIVE RESULTS

We present extensive qualitative results below. We compare FarsightAlign TTS against competing TTS methods on Geneval benchmark (Figures 14 and 15) and DrawBench (Figures 16 and 17). Across these scenarios, our method consistently selects candidates that exhibit superior semantic alignment.

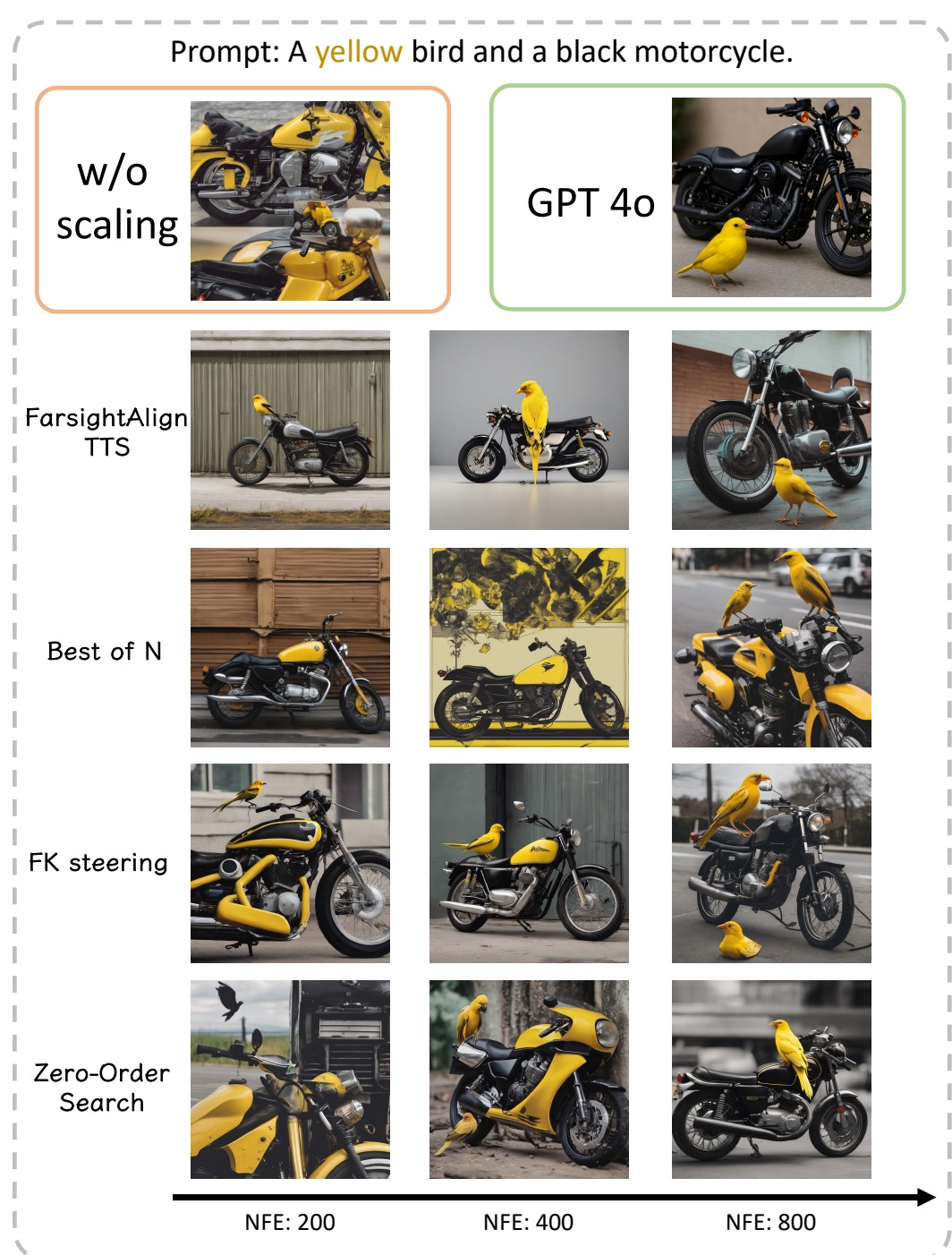

Figure 14: Qualitative comparison on the Geneval benchmark.

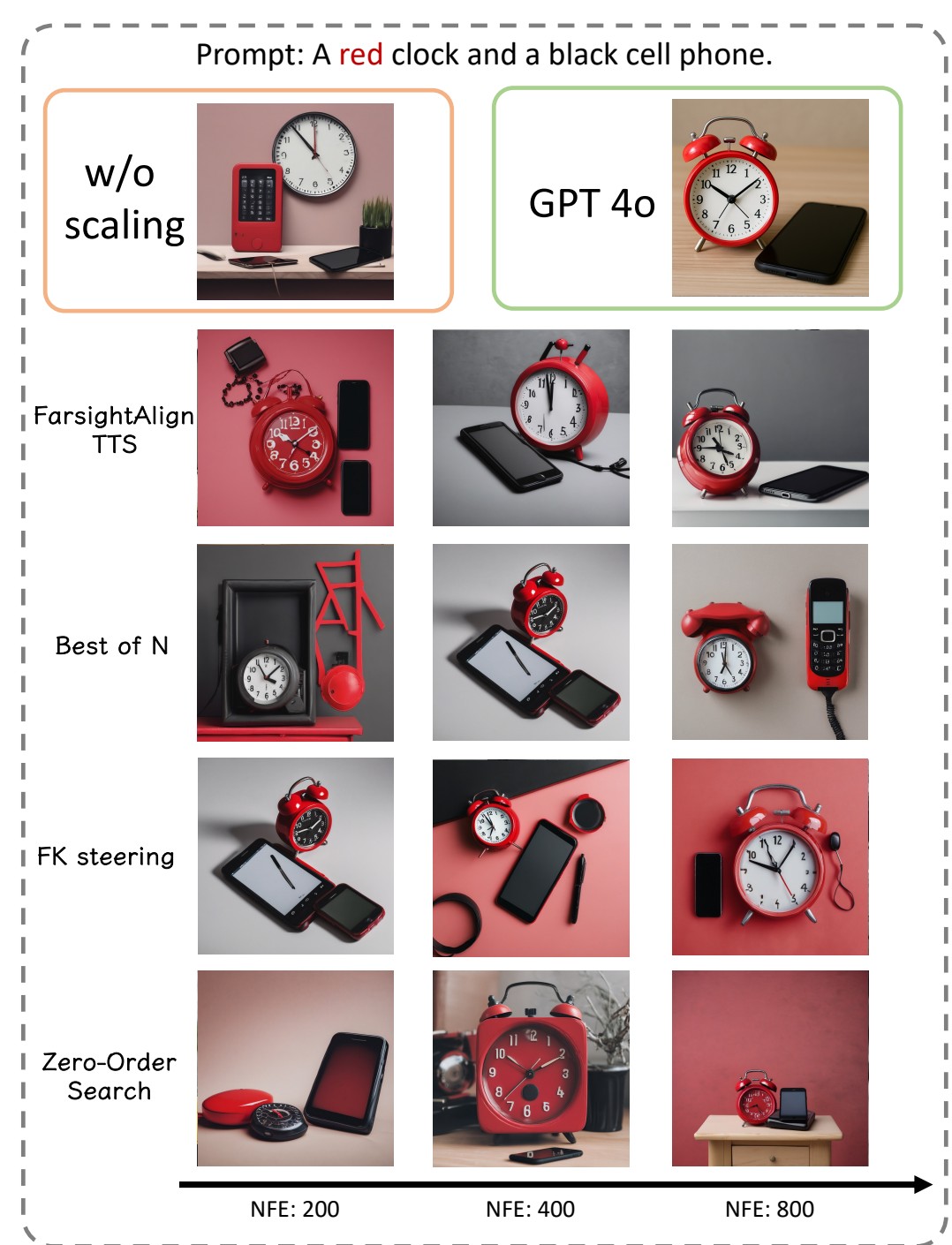

Figure 15: Qualitative comparison on the Geneval benchmark.

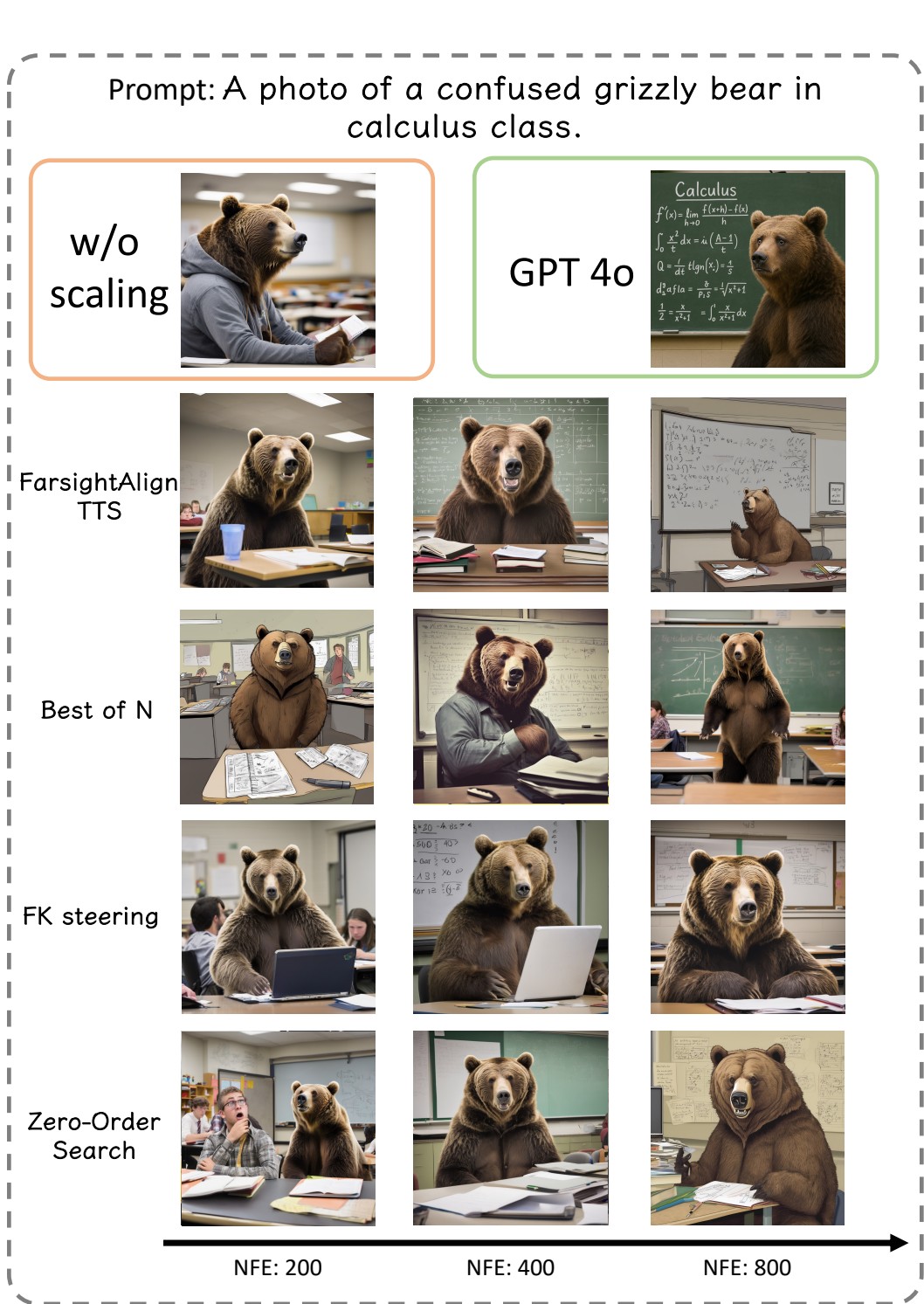

Figure 16: Qualitative comparison on the DrawBench benchmark.

Figure 17: Qualitative comparison on the DrawBench benchmark.

