# OpenReview forum: "FarsightAlign: Early-Stage Test-Time Scaling for Prompt-Aligned Text-to-Image Generation"
_ICLR.cc/2026/Conference — ICLR 2026 Conference Withdrawn Submission_

### Official Review · Reviewer_BtqB · 2025-10-28

**Soundness:** 3
**Presentation:** 3
**Contribution:** 3
**Rating:** 4
**Confidence:** 3

**Summary:**

This paper introduces FarsightAlign TTS, an efficient test-time scaling method to improve how well text-to-image diffusion models follow complex prompts. The authors observe that while early-stage decoded images are blurry and uninformative, the cross-attention maps from the same initial steps already contain a clear semantic blueprint of the final image, including object layout and attributes. Leveraging this insight, FarsightAlign TTS runs multiple candidates for just a few denoising steps, extracts structured semantic information from their attention maps, and uses a lightweight LLM scorer to prune unaligned candidates before committing to a full generation. This early-pruning approach significantly reduces computational cost while outperforming existing methods in prompt alignment, and can also be used as a plug-and-play module to boost other frameworks.

**Strengths:**

1. The paper is very well-written and logically structured, making the core ideas easy to understand. The figures are clear and effectively illustrate the main concepts.

2. The motivation is straightforward and reasonable.

**Weaknesses:**

1. The paper effectively demonstrates semantic extraction for prompts with a simple structure, such as conjoined noun phrases with single adjectives. However, it is unclear how the proposed method scales to prompts with greater syntactic complexity and descriptive depth. For example, in a prompt like, "A thoughtful scientist in a white lab coat is examining a glowing blue flask, while in the background, a complex diagram is faintly visible on a chalkboard," the framework's ability to parse nested relationships, abstract attributes ("thoughtful"), and contextual clauses ("while in the background") is not evaluated. The method's reliance on extracting simple object, attribute, and position triplets may be insufficient for these nuanced scenarios.

2. A significant gap in the evaluation is the lack of head-to-head comparisons with other methods on key benchmarks. The paper should include results on GenEval, T2I-CompBench, and WISE, as their absence makes it difficult to gauge the true effectiveness of this work against existing techniques.

3. The experiments are conducted on SDXL and SD3, which are strong but no longer represent the cutting edge of text-to-image generation. Newer architectures like FLUX.1-dev or Qwen-Image have demonstrated significantly improved native prompt understanding. The critical question is whether FarsightAlign provides a meaningful boost to these already powerful models, or if its benefits are most pronounced on older architectures with known compositional weaknesses. Without testing on these stronger baselines, it is difficult to assess if the proposed method is a universally beneficial tool or a remedial technique for weaker models.

**Questions:**

N/A

---

### Official Review · Reviewer_WJZ9 · 2025-10-31

**Soundness:** 2
**Presentation:** 2
**Contribution:** 2
**Rating:** 2
**Confidence:** 4

**Summary:**

The paper proposes to measure three metrics on the attention map instead of the decoded images during early diffusion steps for T2I TTS tasks. An LLM scorer is employed to score each candidate. The proposed method reduces the computational cost and improves alignment with prompts. The authors have conducted extensive experiments to illustrate the method's effectiveness.

**Strengths:**

1. The idea of using two normal distributions to dynamically determine the threshold to segment the attention map is novel.
2. The figure is clear, and the paper is easy to follow.

**Weaknesses:**

1. Lack of related work. Many earlier works adopt similar methods to apply TTS for T2I tasks like [1]. Authors should discuss the difference between the proposed method and [1]. Besides, the use of a cross-attention map to determine the alignment between the prompts and images has also been widely discussed [2, 3, 4]; the authors provide little information about this topic. The authors are expected to include these references and discuss the connections.
2. A major concern is whether the metrics on the attention map could represent the decoded latents. For example, is a large attention score equal to the object's existence in the image?
3. What is the purpose of using an LLM as a scorer? It seems like a rule-based scoring system could sufficiently conduct scoring for each latent.
4. The definition of the decode loss is not clear. How do the authors obtain the object’s bounding boxes from the GT image and the image decoded from the early latent?
5. A table should be presented for the main metrics when comparing the proposed method with others. This could provide results with more accurate numbers.
6. In Figure 3, the term “confidence” is misspelled as “cofidence.”

[1] Guo, Ziyu, et al. "Can We Generate Images with CoT? Let's Verify and Reinforce Image Generation Step by Step." arXiv preprint arXiv:2501.13926 (2025).

[2] Chefer, Hila, et al. "Attend-and-Excite: Attention-Based Semantic Guidance for Text-to-Image Diffusion Models." ACM Transactions on Graphics 42.4 (2023): 1-10.

[3] Wang, Zirui, et al. "TokenCompose: Text-to-Image Diffusion with Token-level Supervision." Proceedings of the IEEE/CVF Conference on Computer Vision and Pattern Recognition (CVPR). 2024: 8553-8564.

[4] Jiang, Dongzhi, et al. "CoMat: Aligning Text-to-Image Diffusion Model with Image-to-Text Concept Matching." Advances in Neural Information Processing Systems (NeurIPS) 37 (2024).

**Questions:**

1. In the left figure in Fig. 5, why does BoN (Clip Score) fall behind the proposed method? It seems that BoN (Clip Score) should be an upper bound when testing on CLIP score.

Please refer to other questions in the weakness part.

---

### Official Review · Reviewer_ankA · 2025-10-31

**Soundness:** 2
**Presentation:** 3
**Contribution:** 2
**Rating:** 4
**Confidence:** 4

**Summary:**

The paper proposes FarsightAlign TTS, a test-time scaling method for text-to-image diffusion that prunes unpromising candidates early by reading semantic signals from cross-attention maps instead of decoding intermediate latents. After sampling many noises, it runs 5 denoising steps, aggregates token-wise attention maps, segments object regions, and derives object confidence, position, and attribute-to-object binding. These structured summaries are scored by a lightweight LLM judge, and only top candidates proceed to full denoising.

**Strengths:**

- Empirically shows attention-based position error is low at early steps vs. decoded latent, which enables pruning unfavorable candidates after only 5 denoising steps.
- Interesting observation that the intensity distribution of an attention map typically exhibits a distinct bimodal structure.

**Weaknesses:**

- Overhead accounting for the LLM judge and the other elements are claimed to be negligible, yet the paper lacks a wall-clock/VRAM report of various components.
- Benchmarks and metrics are appropriate, but no confidence intervals/significance tests or human studies on semantic alignment are reported.
- The paper provides a well-executed engineering contribution with clear empirical value, but presents no conceptual innovation beyond integration of known components (early cross-attention analysis, LLM-based judging, etc.).

**Questions:**

1. Report per-prompt runtime/VRAM all components in the propoesd pipeline, and compare to decoding-based TTS at matched NFE.
2. How do you select object/attribute tokens (multi-token nouns/adj., synonyms, plural forms)? Any NLP parser/heuristics? Please clarify failures and provide robustness analyses.
3. Add CIs/paired tests for GenEval and ImageReward; per-category breakdowns (attributes/relations/spatial/multi-object) would show where gains arise.
4. Could non-LLM or much smaller heuristic scorers approximate your LLM judge (e.g., rule-based checks on the JSON files), to further cut overhead? You ablate across LLMs; can you include a non-LLM baseline as well?

---

### Official Review · Reviewer_g3Xi · 2025-11-01

**Soundness:** 3
**Presentation:** 4
**Contribution:** 3
**Rating:** 6
**Confidence:** 4

**Summary:**

This paper proposes FarsightAlign TTS, a test-time scaling method designed to improve semantic alignment in text-to-image diffusion models. The approach avoids full-image decoding by extracting semantic cues (e.g., object presence, layout) from early-stage cross-attention maps. These cues are then used by a lightweight LLM scorer to efficiently prune candidate images.The authors report that FarsightAlign TTS outperforms existing test-time scaling methods, particularly in low-computation scenarios, and can be used as a modular plug-in with minimal overhead.

**Strengths:**

1.The paper is well-organized and easy to follow.

2.The core idea of FarsightAlign is simple yet effective. Unlike other TTS methods that score at the full-image level, FarsightAlign's approach of extracting cues from early-stage cross-attention maps is more computationally efficient and appears to be robust.

3.Based on both quantitative and qualitative results, the proposed FarsightAlign achieves better performance than previous methods.

**Weaknesses:**

1.Limited Attribute Handling: The paper provides weak evidence that FarsightAlign can handle complex attributes. It is unclear if the method works for actions (e.g., "running," "waving") or is limited to simple attributes like color. The qualitative results in Figure 16 are not promising, showing an undesirable style shift (to 'comics style') rather than a targeted edit.

2.Limitations of Coarse Cross-Attention: The method's reliance on early-stage cross-attention maps is a potential flaw. These maps are coarse, capturing only approximate shape and layout while lacking fine-grained semantic detail. This could lead to semantic ambiguity; for example, the model may be unable to distinguish between a 'tiger' and a 'lion' if their shapes are similar. I recommend the author to show more cases like that.

**Questions:**

From figure 14 to figure 17, please specify the T2I base model and the corresponding random seed for generating the w/o scaling image.

---

### Note · Authors · 2025-11-13

I have read and agree with the venue's withdrawal policy on behalf of myself and my co-authors.